# Beyond Biochemical Markers: Characterizing Malnutrition in COVID-19

**DOI:** 10.3390/nu18010075

**Published:** 2025-12-26

**Authors:** Katarzyna Plewka-Barcik, Maria Różańska-Trzepla, Krzysztof Kłos, Marta Krawczyk, Andrzej Chciałowski, Stanisław Niemczyk, Anna Matyjek

**Affiliations:** 1Department of Internal Medicine, Infectious Diseases and Allergology, Military Institute of Medicine—National Research Institute, 04-141 Warsaw, Poland; kklos@wim.mil.pl (K.K.); achcialowski@wim.mil.pl (A.C.); 2Department of Internal Medicine, Nephrology and Dialysis Therapy, Military Institute of Medicine—National Research Institute, 04-141 Warsaw, Poland; mrozanska@wim.mil.pl (M.R.-T.); sniemczyk@wim.mil.pl (S.N.); amatyjek@wim.mil.pl (A.M.); 3Department of Immunology, Mossakowski Medical Research Institute, Polish Academy of Sciences, 02-106 Warsaw, Poland; mkrawczyk@imdik.pan.pl; 4Doctoral School of Translational Medicine, Mossakowski Medical Research Institute, Polish Academy of Sciences, Centre of Postgraduate Medical Education, 02-106 Warsaw, Poland; 5Clinical Trials Support Unit, Medical University of Lodz, 92-213 Lodz, Poland

**Keywords:** COVID-19, SARS-CoV-2, malnutrition, body composition

## Abstract

**Background/Objective**: Malnutrition is common in hospitalized patients and worsens clinical outcomes, particularly in coronavirus disease 2019 (COVID-19), in which inflammation and metabolic disruption contribute to nutritional decline. Thus, identifying simple and accessible markers is essential for early detection and intervention to prevent further deterioration. This study aimed to investigate biochemical and body composition changes during COVID-19 hospitalization and identify key features of hospital-acquired nutritional status disorders. **Methods**: We conducted a prospective, single-center, observational study of 66 patients hospitalized with COVID-19 between December 2020 and June 2021. Biochemical markers and body composition parameters were measured at admission and at discharge. Deterioration of nutritional status was defined as a weight loss of more than 3% during hospitalization. **Results**: A total of 66 patients (61% male, aged 56.7 ± 13.4 years; 39% female, aged 58.8 ± 12.0 years) were included. Deterioration of nutritional status was observed in 20 (30%) individuals, more likely in men (OR 7.94, 95% CI: 1.28–49.08) and patients with longer hospitalization (OR 1.30 per day, 95% CI: 1.08–1.57). Weight loss was primarily characterized by a reduction in adipose tissue mass, whereas lean tissue mass did not change significantly. Traditional biochemical markers of malnutrition, including low albumin, prealbumin, or cholesterol levels, were not present in this cohort. **Conclusions**: Our study highlights the significant burden of nutritional deterioration in hospitalized patients with COVID-19 and demonstrates its atypical presentation, which may limit the effectiveness of standard malnutrition assessment tools.

## 1. Introduction

Malnutrition among hospitalized patients is a significant concern and has been associated with unfavorable clinical outcomes, including longer hospital stays, higher complication rates, and increased mortality [1,2]. During the coronavirus 2019 (COVID-19) pandemic, nutritional disturbances emerged as an important yet often underestimated determinant of prognosis, affecting both hospitalized and nonhospitalized patients [3].

Beyond respiratory involvement, severe acute respiratory syndrome coronavirus 2 (SARS-CoV-2) induces systemic inflammation and profound metabolic dysregulation [4]. These systemic disturbances promote tissue catabolism, attenuate immune responses, and contribute to poor clinical outcomes. Adequate nutritional status plays a fundamental role in modulating the immune response; micronutrient deficiencies and protein–energy undernutrition compromise both innate and adaptive immune responses, increasing susceptibility to infections and worsening their course [5,6,7]. In COVID-19, malnutrition and immune dysfunction may reinforce one another: limited nutritional reserves reduce immune competence, whereas the hyperinflammatory state intensifies metabolic demands and promotes further nutritional decline [8].

Importantly, nutritional deterioration in COVID-19 may also be driven by disease-specific symptoms that affect nutritional intake. Gastrointestinal manifestations, such as nausea, vomiting, diarrhea, and abdominal pain, are frequently reported and can reduce both appetite and nutrient absorption [9,10]. Chemosensory disturbances, particularly anosmia and dysgeusia, are also common and further compromise dietary adequacy [11].

To guide clinical practice, the European Society for Clinical Nutrition and Metabolism (ESPEN) defines malnutrition as a condition resulting from reduced nutrient intake or uptake, accompanied by adverse changes in body composition, most notably unintentional weight loss, low body mass index (BMI), or diminished fat-free mass. These alterations ultimately compromise the physiological and/or mental functions. ESPEN proposes a two-option diagnostic framework: malnutrition may be diagnosed either by a BMI below 18.5 kg/m^2^ or by the presence of unintentional weight loss (>10% at any time or >5% within three months) combined with a low age-adjusted BMI (<20 kg/m^2^ for individuals < 70 years; <22 kg/m^2^ for those ≥70 years) or a reduced fat-free mass index (FFMI < 15 kg/m^2^ in women; <17 kg/m^2^ in men) [12]. Complementing ESPEN’s position, the Global Leadership Initiative on Malnutrition (GLIM) recommends a diagnostic framework that combines phenotypic (weight loss, low BMI, and reduced muscle mass) and etiological criteria (inflammation, reduced intake, or assimilation) [13]. Although conceptually robust, both frameworks may be difficult to apply in acute infectious conditions such as COVID-19, where metabolic deterioration can occur over days rather than weeks or months.

Traditional laboratory markers have also been used to support nutritional evaluation; however, their diagnostic value appears to be limited in the context of acute illness. Albumin, prealbumin, and transferrin are strongly influenced by inflammation, fluid shifts, and disease severity, which reduces their specificity for malnutrition. Additional biomarkers, such as serum creatinine, total cholesterol, hemoglobin, and lymphocyte counts, may offer complementary insights but similarly reflect metabolic stress rather than nutritional decline. As highlighted by Keller et al., these parameters are more indicative of inflammatory burden than of true nutritional status [14].

In contrast to biochemical markers, the assessment of body composition may offer a more direct and sensitive means of characterizing early nutritional deterioration in COVID-19. Research on critically ill patients has demonstrated substantial loss of fat-free mass and skeletal muscle, often linked to disease severity [15]. However, evidence regarding body composition changes in moderately ill patients or those requiring sub-intensive care remains limited. Bioimpedance spectroscopy (BIS) is a noninvasive and objective method for assessing body composition and hydration status and is widely used in dialysis patients for routine evaluation of both parameters. Its ease of use at the bedside makes it a practical tool for monitoring changes in nutritional and fluid status in patients with COVID-19 [16,17]. Nevertheless, the diagnostic utility of BIS-derived parameters in moderately ill patients has not yet been well established.

Given the limitations of standard diagnostic tools for malnutrition in patients with COVID-19 and considering the dynamic metabolic changes observed in this population, we conducted a study to evaluate the prevalence of nutritional status deterioration in patients hospitalized for COVID-19, and to identify the key disturbances in traditional laboratory markers of malnutrition and body composition in this population.

## 2. Materials and Methods

### 2.1. Study Design and Participants

This was a prospective, observational, single-center study embedded within the CoVTE (Coronavirus 2019-associated Venous Thromboembolism) study framework (data unpublished yet). Patients were recruited between December 2020 and June 2021 from the COVID-19 units of the Military Institute of Medicine in Warsaw, Poland. The inclusion criteria were age ≥18 years, symptomatic SARS-CoV-2 infection confirmed by real-time reverse transcriptase-polymerase chain reaction (RT-PCR) testing of nasopharyngeal swabs at admission, preserved functional capacity, and the ability to provide informed consent. The exclusion criteria were contraindications to bioimpedance spectroscopy, nephrotic-range proteinuria, and inability or unwillingness to participate in the study. This study was approved by the Ethics Committee of the Military Institute of Medicine (number 2/WIM/2021). Written informed consent was obtained from all participants.

### 2.2. Data Collection

Nutritional status was evaluated as part of the ancillary analysis of the CoVTE study. Nutritional parameters were assessed at admission and discharge from the COVID-19 unit. The following measures were included: concentration of serum albumin, prealbumin, lipid profile, phosphorus, uric acid, routine hematologic and renal function parameters and body composition assessed using bioimpedance spectroscopy (BIS). Body composition was assessed using a multi-frequency bioimpedance device (Body Composition Monitor, Fresenius Medical Care, Bad Homburg, Germany). The parameters used—adipose tissue mass (ATM), lean tissue mass (LTM), and overhydration (OH)—are part of the standard set of output variables provided by the body composition monitor device, derived from validated multifrequency bioimpedance spectroscopy algorithms. All body composition measurements were conducted by trained medical personnel with patients in the supine position, before noon and prior to their main meal, following a short rest and bladder voiding. The same standardized protocol was applied at both time points to ensure the consistency of the results. Body weight and height were measured at admission using a calibrated electronic scale and a wall-mounted stadiometer (Charder MS4900, Charder Electronic Co., Ltd., Taichung, Taiwan), with the patients in light clothing and without shoes. Weight was recorded before noon and prior to the main meal and reassessed at discharge under the same conditions. Medical history and clinical data on the course of COVID-19 were also collected.

Deterioration of nutritional status was defined as a weight loss of more than 3% during hospital stay.

### 2.3. Statistical Analysis

Quantitative variables were expressed as counts and percentages and compared using the chi-square test with Yates’ correction or Fisher’s exact test. Continuous variables were tested for normal distribution (Shapiro–Wilk test) and reported as mean ± SD or median (interquartile range, IQR), as appropriate. Between-group and within-group comparisons were performed using t-tests or their nonparametric equivalents.

Deterioration of nutritional status was defined as a weight loss of >3% during hospitalization. Key characteristic features associated with this state were identified using univariate and multivariable logistic regression, with variable selection performed using stepwise and Least Absolute Shrinkage and Selection Operator (LASSO) methods. The quality of the model was evaluated using area under the curve (AUC) with 95% confidence intervals (CI) and visualized using the Receiver Operating Characteristic (ROC) curve. Analyses were conducted using Statistica version 13.3 (TIBCO Software, Palo Alto, CA, USA), with *p* < 0.05 considered significant. Missing data (<8%) were imputed using the k-nearest-neighbor algorithm.

### 2.4. Integration of Generative Artificial Intelligence (GenAI)

Generative AI was not used to design, conduct or analyze the data of this study.

## 3. Results

Among the 114 participants in the CoVTE study, 66 with complete nutritional evaluation data were included in this analysis. The final study cohort included 40 men (61%) and 26 women (39%), with a mean age of 58 ± 13 years. The most common comorbidities were hypertension (38%), diabetes mellitus (12%), chronic respiratory disease (12%), coronary artery disease (11%), and neoplasia (8%). A total of 57 patients required oxygen therapy. Consequently, systemic intravenous glucocorticoid therapy was administered to 52 patients in accordance with the clinical guidelines for the management of COVID-19–associated pneumonia with severe respiratory failure.

At admission, none of the participants had a BMI below 18.5 kg/m^2^; at discharge, a BMI below this threshold was identified in one patient.

Deterioration of nutritional status during hospital stay was observed in 20 of 66 (30%) patients. Among them, 11 individuals (17% of all patients) experienced weight loss of more than 5%—meeting the key criterion for malnutrition.

Patients with deterioration of nutritional status were more frequently male (85% vs. 50%); the other demographic and anthropometric parameters did not differ between the groups (Table 1). Among the baseline biochemical parameters, only the prealbumin was found to be significantly lower in the group with deterioration of nutritional status (Table 1).

### 3.1. Changes in Nutritional Parameters During Hospitalization

The median duration of hospital stay was 13 days (IQR: 10–15 days, range: 5–36 days).

Table 2 summarizes the biochemical and body composition parameters at admission and discharge of the patients. No reductions were observed in laboratory parameters traditionally considered markers of malnutrition: serum albumin, prealbumin, and total cholesterol. In contrast, both prealbumin (+15.5 ± 8.8 mg/dL, *p* < 0.001) and total cholesterol (+25.0 ± 34.8 mg/dL, *p* < 0.001) concentrations increased significantly. Similarly, there was an increase in phosphorus (+0.2 ± 0.8 mg/dL, *p* = 0.034) and uric acid (+0.6 mg/dL, *p* = 0.058) levels—biochemical parameters reflecting catabolic process. Kidney function slightly improved, while hemoglobin level decreased during hospital stay, likely reflecting resolution of dehydration.

No significant changes were detected in the lean tissue mass or lean tissue index, and hydration-related measures. Fat-related measures showed modest yet statistically significant reductions (fat mass, –0.5 kg; adipose tissue mass, –0.6 kg; both *p* ≈ 0.03), which appeared to be the most noticeable trend within the body composition parameters.

### 3.2. Univariate Analysis

In univariate regression analysis, male sex, adipose tissue mass (ATM) reduction, and phosphorus concentration decrease were significantly associated with deterioration of nutritional status. Also, the duration of the stay at the COVID unit seemed to be an important factor, as the result of the analysis was close to significance (*p* = 0.055) (Table 3).

### 3.3. Multivariable Analysis

In multivariable logistic regression (Table 4), three factors were independently associated with the occurrence of nutritional deterioration: male sex (OR 7.94; 95% CI: 1.28–49.08; *p* = 0.026), COVID unit stay duration (OR 1.30; 95% CI: 1.08–1.57; *p* = 0.005), and a greater loss of adipose tissue mass (ATM; OR 0.80; 95% CI: 0.69–0.94; *p* = 0.006).

This model showed a high-quality effect in describing patients’ experiences of deterioration of nutritional status during hospital stay—demonstrating AUC 0.85 (95% CI: 0.76–0.94) (Figure 1).

## 4. Discussion

In our study, deterioration of nutritional status, defined as a body weight reduction of >3% during hospitalization, was observed in 30% of patients, including 17% who lost >5% of their baseline weight, with weight loss being more common in men. Weight loss during acute COVID-19 has been reported relatively frequently and is often linked to inflammation, reduced food intake, and hospitalization-related stress. Di Filippo et al. documented a median weight change of −3.0% between admission and one month after discharge, with 57.8% of patients losing >2% and 35.1% losing >5% of their pre-admission weight [18]. Kröönström et al. found that 77.5% of men and 58.8% of women reported weight loss during infection, with a significant sex difference [19]. Larger epidemiological studies further support the high burden of nutritional impairment in COVID-19: a meta-analysis by Boaz et al. involving 354,332 survivors indicated that more than half of hospitalized adults were malnourished or at risk, while Abate et al. and Feng et al. reported prevalence estimates of 49.1% and 70.7%, respectively [20,21,22]. Our findings are consistent with the literature and indicate that measurable nutritional decline may also occur in patients with relatively short hospital stays and moderate or sub-intensive COVID-19 severity. Comparable observations have been reported in acute non-COVID-19 conditions: Braunschweig et al. documented in-hospital nutritional deterioration in 31% of adults, while Botero et al. reported a worsening nutritional status in 23.8% of hospitalized patients [23,24].

We found men and patients with longer hospital stays as more likely to develop nutritional disorders. This tendency is broadly consistent with prior reports indicating that prolonged hospital stays may contribute to worsening nutritional status [25], while other studies have shown that malnutrition can also occur rapidly, irrespective of hospitalization duration [22] and may vary depending on care intensity [26,27]. The sex-related patterns observed in our cohort may align with the findings of Tosato et al. In their study, no initial sex differences were observed in the overall prevalence of malnutrition among COVID-19 survivors (23% in men vs. 21% in women); however, a notable difference emerged among participants aged >65 years (30% vs. 17%). After multivariable adjustment, male sex remained an independent risk factor (OR 5.56, 95% CI 3.53–8.74) [27]. Although these observations do not establish causality, they may suggest that men could be more vulnerable to nutritional decline during acute COVID-19, a hypothesis that warrants further investigation.

A noteworthy observation is that weight loss in our cohort was characterized primarily by reductions in adipose rather than lean tissue, as assessed by bioimpedance spectroscopy (BIS), which may detect changes not captured by conventional biochemical markers. The lean tissue mass and hydration indices remained stable, whereas the adipose tissue mass decreased significantly. This pattern differs from the classical trajectories of acute malnutrition, in which early lean tissue catabolism is often anticipated, and may be related to the relatively short median hospital stay and metabolic features of acute COVID-19. Accelerated fat mobilization has been described in COVID-19 and may contribute to inflammatory and metabolic stress [28,29,30]. While moderate fat utilization may represent an adaptive response to acute illness, more pronounced adipose tissue loss could indicate metabolic strain rather than a fully effective adaptive response. These interpretations remain speculative and warrant confirmation in larger prospective cohorts, particularly as most available data on body composition are derived from critically ill populations, where lean tissue loss is more pronounced [31].

The dynamics of biochemical indices in our cohort further illustrate the complex relationship between laboratory markers and nutritional status during acute infection. Albumin, total protein, and prealbumin levels did not decline; instead, prealbumin and cholesterol levels increased during hospitalization. Although these trends diverge from classical protein–energy malnutrition, they likely reflect the limited specificity of biochemical markers under systemic inflammation, as both albumin and prealbumin are negative acute-phase reactants influenced more by inflammatory signaling, vascular permeability, and fluid shifts than by short-term nutritional intake. Similarly, increased cholesterol levels may be related to systemic glucocorticoid therapy rather than reflecting improved nutritional reserves. Together, these observations underscore the need for cautious interpretation of biochemical markers when assessing the nutritional status of acutely ill patients.

Changes in the phosphorus and uric acid concentrations were also observed. Phosphorus homeostasis during acute illness is highly dynamic and affected by inflammation, renal function, metabolic shifts, and catabolism. As noted by Anghel et al., COVID-19 may alter intracellular phosphorus distribution through mechanisms such as respiratory alkalosis, inadequate nutritional intake, and renal impairment [32]. In our cohort, the absence of concurrent lean tissue loss and the improvement in renal function, reflected by higher eGFR and lower creatinine levels, most likely due to intravenous fluid administration, suggest that fluctuations in phosphorus and uric acid should be interpreted with caution. Importantly, no features consistent with refeeding syndrome were observed, despite the potential vulnerability of malnourished patients to such disturbances during early nutritional recovery.

Overall, our findings suggest that nutritional deterioration in patients with moderate and sub-intensive COVID-19 is characterized predominantly by reductions in adipose tissue, with relative preservation of lean tissue and without the biochemical profile typically associated with malnutrition. This pattern differs from that of chronic or prolonged critical illnesses and highlights the importance of integrating compositional methods, such as BIS, with clinical assessment rather than relying solely on biochemical indices. Furthermore, the higher occurrence of nutritional decline among men and patients with longer hospitalization underscores the relevance of individual clinical characteristics in shaping the trajectory of nutritional status during acute COVID-19.

The limitations of this study include the single-center setting, small sample size, and absence of post-discharge follow-up. Additionally, standardized nutritional screening was not applied, which may limit the comparability with other studies. Another limitation is the use of a non-standard definition of malnutrition based on >3% weight loss during hospitalization. This threshold was selected pragmatically because of a short observation window. Our definition aimed to capture early and potentially meaningful deterioration during acute illness but may limit the direct comparability with studies employing established diagnostic frameworks.

## 5. Conclusions

Our findings indicate that deterioration of nutritional status in patients hospitalized for COVID-19 is more frequently seen in men and in those with prolonged hospital stay, and is characterized predominantly by adipose tissue rather than lean tissue loss, while biochemical markers appear to have limited usefulness. These results suggest that standard nutrition assessment tools may have reduced sensitivity in the context of acute infectious diseases such as COVID-19.

## Figures and Tables

**Figure 1 nutrients-18-00075-f001:**
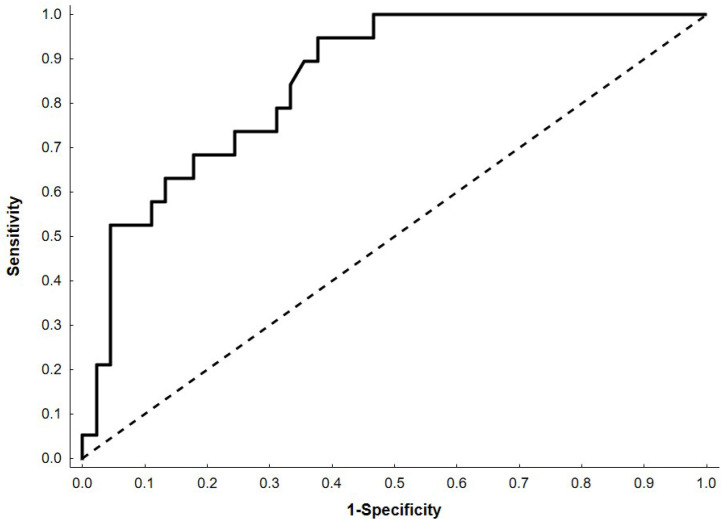
Receiver Operating Characteristic (ROC) curve for the multivariable logistic regression model. The black dashed line represents the line of no discrimination.

**Table 1 nutrients-18-00075-t001:** Baseline characteristics of the study group population.

Parameter	All Patients (*N* = 66)	Deterioration of Nutritional Status (Weight Loss of >3%) (*N* = 20)	No Deterioration (*N* = 46)	*p*-Value
**Demographic and Anthropometric Parameters**				
Male sex, *n* (%)	40 (61%)	17 (85%)	23 (50%)	0.016
Age [years]	58 ± 13	59.0 ± 14.1	56.8 ± 12.4	0.55
Height [cm]	172.5 ± 9.5	175.2 ± 8.5	172.2 ± 9.9	0.22
Weight baseline [kg]	85 ± 19	85.3 ± 15.8	87.4 ± 20.4	0.65
BMI baseline [kg/m^2^]	27.6 (25.2–31.6)	27.1 (25.3–30.4)	27.8 (25.2–32.3)	0.494
**Comorbidities**				
Hypertension, *n* (%)	25 (38%)	8 (40%)	17 (37%)	0.815
Coronary artery disease, *n* (%)	7 (11%)	2 (10%)	5 (11%)	0.742
Diabetes, *n* (%)	8 (12%)	1 (5%)	7 (15%)	0.448
Respiratory system diseases, *n* (%)	8 (12%)	3 (15%)	5 (11%)	0.950
Neoplasm, *n* (%)	5 (8%)	2 (10%)	3 (7%)	0.988
**Baseline Biochemical Parameters**				
Hemoglobin [g/dL]	13.9 ± 1.8	13.8 ± 1.4	13.9 ± 1.9	0.759
Creatinine [mg/dL]	0.9 (0.7–1.1)	0.9 (0.8–1.1)	0.9 (0.7–1.1)	0.743
Urea [mg/dL]	30 (22–36)	30.5 (26–36.5)	30 (21–36)	0.655
eGFR [mL/min/1.73 m^2^]	85 ± 21.6	91.8 ± 22.8	82 ± 20.6	0.900
Serum albumin [g/dL]	3.6 ± 0.4	3.5 ± 0.4	3.6 ± 0.3	0.131
Phosphorus [mg/dL]	3.3 ± 0.7	3.3 ± 0.7	3.3 ± 0.7	0.693
Uric acid [mg/dL]	4.3 ± 1.3	4.5 ± 1.2	4.2 ± 1.3	0.477
Total cholesterol [mg/dL]	139.6 ± 34.3	135.5 ± 35.9	141.4 ± 33.9	0.525
LDL [mg/dL]	81.5 ± 30.3	77.3 ± 29.4	83.3 ± 30.7	0.456
HDL [mg/dL]	30.5 (27–40)	29.5 (27.5–40)	31.5 (27–39)	0.867
Triglycerides [mg/dL]	116 (94–143)	115 (79.5–131.5)	116 (97–148)	0.596
Prealbumin [mg/dL]	12.3 (9–15.9)	10.5 (7.6–12.3)	13.6 (10.3–17.3)	0.009

eGFR—estimated glomerular filtration rate; LDL—low-density lipoprotein; HDL—high-density lipoprotein.

**Table 2 nutrients-18-00075-t002:** Biochemical and body composition parameters of COVID-19 patients.

Parameter	Baseline	End	Change (∆)	*p*-Value
**Biochemical Parameters**				
Hemoglobin [g/dL]	13.9 ± 1.8	13.2 ± 1.5	−0.7 ± 1.3	<0.001
Creatinine [mg/dL]	0.9 (0.7–1.1)	0.8 (0.7–0.9)	−0.1 (−0.2–0)	<0.001
Urea [mg/dL]	30 (22–36)	29 (23–37)	2 (−7–7)	0.965
eGFR [mL/min/1.73 m^2^]	83 (73–101)	95 (83–110)	12 (0–25)	<0.001
Serum albumin [g/dL]	3.6 ± 0.4	3.6 ± 0.4	0.03 ± 0.39	0.583
Total protein [g/dL]	6.2 ± 0.6	6.2 ± 0.6	−0.04 ± 0.67	0.639
Phosphorus [mg/dL]	3.3 ± 0.7	3.5 ± 0.5	0.2 ± 0.8	0.034
Uric acid [mg/dL]	4.2 (3.4–5.2)	4.6 (3.8–5.4)	0.6 (−0.7–1.5)	0.058
Total cholesterol [mg/dL]	139.6 ± 34.3	164.6 ± 36.8	25.0 ± 34.8	<0.001
LDL [mg/dL]	80.5 ± 30.9	99.9 ± 30.6	19.4 ± 28.2	<0.001
HDL [mg/dL]	34.3 ± 10.4	37.8 ± 9.4	3.5 ± 9.8	0.008
Triglycerides [mg/dL]	116 (95.5–143.5)	169.5 (124.5–135.5)	52.0 (16.5–91)	<0.001
Prealbumin [mg/dL]	12.2 ± 4.8	27.8 ± 8.2	15.5 ± 8.8	<0.001
**Body Composition Parameters**				
Weight [kg]	85.0 (73.0–98.8)	84.5 (71.8–97.3)	−0.95 (−2.8–0.2)	<0.001
BMI [kg/m^2^]	27.6 (25.2–31.6)	26.6 (24.8–31.3)	−0.3 (−1–0)	<0.001
LTM [kg]	37 (29–46)	37.7 (28.6–46.5)	−0.3 (−2.7–2.7)	0.784
Percentage of lean tissue [%]	45.0 (37.0–53.6)	47.9 (35.8–53.6)	0.2 (−2.9–4.4)	0.342
Lean tissue index (LTI) [kg/m^2^]	12.6 (10.5–14.2)	12.7 (10.4–14.5)	−0.1 (−1.0–0.9)	0.825
LTI difference to reference [kg/m^2^]	−0.7 (−1.6–1.0)	−0.9 (−1.7–0.9)	−0.1 (−1.0–0.9)	0.846
ATM [kg]	45.6 (37.6–55.4)	42.9 (34.2–55.9)	−0.6 (−4.0–1.6)	0.028
Fat tissue index (FTI) [kg/m^2^]	15.2 (12.0–18.2)	14.7 (11.3–17.2)	−0.2 (−1.5–0.5)	0.04
FTI difference to reference [kg/m^2^]	9.2 (6.7–12.3)	9.4 (5.4–12.6)	−0.2 (−1.5–0.5)	0.038
BCM [kg]	19.6 (14.8–25.5)	20.5 (14.5–25.8)	−0.6 (−1.9–2.2)	0.824
OH [L]	0.45 (−0.6–1.3)	0.45 (−0.7–1.5)	0.2 (−0.9–1.0)	0.527

Δ—change calculated as the difference between the end-of-hospitalization and baseline values (Δ = End—Baseline). Negative values indicated a decrease during hospitalization, whereas positive values indicated an increase. Changes are presented as mean ± SD or median (IQR), depending on data distribution; BMI—body mass index; LTM—lean tissue mass; ATM—adipose tissue mass; BCM—body cell mass; OH—overhydration.

**Table 3 nutrients-18-00075-t003:** Univariate logistic regression analysis of factors associated with hospital-acquired malnutrition in patients with COVID-19.

Parameter	OR (95% CI)	*p*-Value
Male sex	5.67 (1.46–22.01)	0.012
Age	1.01 (0.97–1.06)	0.520
Symptoms duration [days]	0.88 (0.76–1.02)	0.087
COVID unit stay [days]	1.13 (1.00–1.28)	0.055
∆ LTM	1.00 (0.90–1.11)	0.988
∆ ATM	0.84 (0.74–0.95)	0.006
∆ Urea	1.01 (0.96–1.06)	0.675
∆ Phosphorus	0.46 (0.22–0.97)	0.041
∆ Serum Albumin	0.68 (0.18–2.60)	0.568
∆ Uric acid	0.72 (0.48–1.07)	0.105
∆ Total Cholesterol	1.00 (0.99–1.02)	0.878
∆ LDL	1.00 (0.98–1.02)	0.997
∆ HDL	1.00 (0.94–1.06)	0.971
∆ Triglycerides	1.001 (0.995–1.007)	0.726
∆ Prealbumin	1.01 (0.95–1.09)	0.681

OR—odds ratio; CI—confidence interval.

**Table 4 nutrients-18-00075-t004:** Multivariable logistic regression model describing hospital-acquired deterioration of nutritional status.

Variable	Coefficient ± SE	OR (95% CI)	*p*-Value
Male sex	2.07 ± 0.93	7.94 (1.28–49.08)	0.026
COVID unit stay (days)	0.27 ± 0.10	1.30 (1.08–1.57)	0.005
Δ ATM	−0.22 ± 0.08	0.80 (0.69–0.94)	0.006

SE—standard error.

## Data Availability

For additional information, please contact the corresponding author. The data are not publicly accessible due to privacy considerations.

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
