# Peer review of "Beyond Biochemical Markers: Characterizing Malnutrition in COVID-19"

_nutrients, 2025, doi:10.3390/nu18010075_

Round 1
Reviewer 1 Report
Comments and Suggestions for Authors
The present study aimed at evaluating nutritional status changes in hospitalized patients with COVID-19.
1) The introduction is very short. Not encompassing the current literature available in malnutrition, hospital malnutrition, and COVID-19-related malnutrition.
2) Line 54: The SGA is not a malnutrition screening tool (risk of malnutrition) but instead a tool for the assessment of malnutrition (diagnosis).
3) Lines 56-57: why are NRS-2002 and SGA not designated to monitor nutritional status overtime? Is that indicated in the original papers?
4) the authors reported changes in body weight and BMI. But no information on body weight and height measurement is provided.
5) Despite this is a study on body composition assessment, no information on how the BIA measurements have been performed is provided.
6) the ATM, LTM and OH are not the parameters commonly adopted for body composition parameters.
7) Was power analysis conducted? it is not provided in the statistical analysis section.
8) How was the “longer” hospital stay defined?
9) the discussion and the conclusion are too “enthusiastic” considering the very limited number of participants.
Author Response
Thank you very much for taking the time to review this manuscript. Please find the detailed responses below and the corresponding revisions in the re-submitted files
Comments 1. The introduction is very short. Not encompassing the current literature available in malnutrition, hospital malnutrition, and COVID-19-related malnutrition.
Response 1. We agree that the original version of the Introduction was overly concise and did not sufficiently reflect the breadth of current evidence on malnutrition in hospitalized patients, including COVID-19-related malnutrition. Therefore, we have revised and expanded this section to better contextualize the study within the existing literature. Specifically, we added a brief overview of the concept of immunonutrition (lines 52–58), discussed COVID-19–related symptoms that may negatively affect nutritional intake (lines 59–63), and incorporated standardized definitions of malnutrition according to ESPEN and GLIM criteria (lines 64–79). Although the initial version of the manuscript briefly addressed the SGA and NRS-2002 screening tools, these descriptions were removed during revision due to their lack of direct application in the present study. These revisions were made to strengthen the scientific background while maintaining consistency with the primary focus of the manuscript.
Comments 2. Line 54: The SGA is not a malnutrition screening tool (risk of malnutrition) but instead a tool for the assessment of malnutrition (diagnosis).
Response 2. In the original version of the manuscript, the SGA was described as a tool for the assessment of a patient’s nutritional status rather than explicitly as a screening instrument. However, following revision of the Introduction, we decided to remove the discussion of the SGA entirely, as its use was not directly related to the study design or to the anthropometric and clinical parameters analyzed in the present work. This modification helped to improve conceptual accuracy and maintain coherence with the scope of the study.
Comments 3. Lines 56-57: why are NRS-2002 and SGA not designated to monitor nutritional status overtime? Is that indicated in the original papers?
Response 3. In the revised version of the manuscript, the discussion of NRS-2002 and SGA was removed, as these tools were not applied in the study methodology and were not used for data analysis or outcome assessment. To maintain coherence between the Introduction and the presented results, we decided not to elaborate on their longitudinal applicability, which is beyond the scope of the current work.
Comments 4. the authors reported changes in body weight and BMI. But no information on body weight and height measurement is provided.
Response 4. We would like to clarify that body weight and height were measured using standard procedures by medical personnel. Weight was assessed with calibrated electronic scales, and height was measured in the upright position using a wall-mounted stadiometer. These methodological details have now been added to the Methods section (lines 131–135). In addition, baseline values of body weight and BMI for the study population have been included in Table 1, while changes in these parameters are presented in Table 2.
Comments 5. Despite this is a study on body composition assessment, no information on how the BIA measurements have been performed is provided.
Response 5. We have now added a detailed description of the bioimpedance measurements to the Methods section. Briefly, body composition was assessed using a multifrequency bioimpedance analyzer (Body Composition Monitor, BCM; Fresenius Medical Care, Bad Homburg, Germany) with patients measured before noon, in a supine position, after voiding and prior to food intake. The procedure was carried out by trained personnel, following the manufacturer’s guidelines (lines 123-131).
Comments 6. the ATM, LTM and OH are not the parameters commonly adopted for body composition parameters.
While we understand that adipose tissue mass, lean tissue mass and hydration status may not represent the most commonly adopted parameters in nutritional research, we would like to clarify that these were not the only body composition variables analyzed in our study, as shown in Table 2. These particular parameters are, however, part of the standard output set generated by the Body Composition Monitor (BCM, Fresenius Medical Care). Although they are not traditionally used as nutritional indicators per se, they have been applied in BIS-based studies, especially in nephrology and hospitalized populations, to assess shifts in tissue and fluid compartments.Additionally, the study was conducted as a collaboration between the Departments of Infectious Diseases and Nephrology, which allowed for combined clinical and technical expertise in the application and interpretation of BIS-derived parameters. We have now clarified this point in the revised Methods section (lines 123-128).
Comments 7. Was power analysis conducted? it is not provided in the statistical analysis section.
A formal power analysis was not conducted, either a priori or post hoc, as this was an exploratory study based on an available patient cohort. We acknowledge this as a limitation and are aware of the methodological considerations related to power estimation. The available sample size may have allowed for the statistical analyses described in the Methods section; however, the findings should be interpreted cautiously.
Comments 8. How was the “longer” hospital stay defined?
Response 8. In our multivariable logistic regression model, hospital stay duration was treated as a continuous variable, expressed in days. The reported odds ratio (OR 1.30) indicates that each additional hospital day was associated with a 30% increase in the odds of malnutrition.
We did not apply a specific categorical threshold (e.g., defining a “longer” stay), as modeling the variable continuously preserved its full informational value and contributed to a better model fit.
Comments 9. the discussion and the conclusion are too “enthusiastic” considering the very limited number of participants.
Response 9. We have carefully revised both the Discussion and Conclusion sections to adopt a more measured and balanced tone. Several statements have been reworded to reflect the exploratory and hypothesis-generating nature of our findings. These adjustments were made to ensure that the interpretation of our results remains proportionate to the study’s scope and statistical power. We are grateful for the reviewer’s feedback, which helped strengthen the clarity and scientific rigor of the manuscript.
Reviewer 2 Report
Comments and Suggestions for Authors
The abstract clearly and logically presents the study’s objectives, methods, and main findings. The topic is timely and clinically relevant, particularly in the context of COVID19, and it effectively highlights the novel approach of body composition based assessment alongside traditional laboratory parameters. The introduction is coherent, well-referenced, and appropriately sets up the study aims, guiding the reader into the manuscript. The methodology is precise and welldocumented. The statistical analysis is comprehensive, spanning from univariate to multivariable models, and clearly identifies male sex, longer hospital stay, and adipose tissue loss as independent predictors of malnutrition. The model performance appears reliable.
Constructive questions:
Does the study follow standardized protocols for body composition measurements, and how reproducible are these measurements across other institutions (eg. using SGA)?
Is there a plan for longer term follow-up to monitor nutritional status post-hospitalization and evaluate the persistence of these effects?
Author Response
Comments 1. Does the study follow standardized protocols for body composition measurements, and how reproducible are these measurements across other institutions (eg. using SGA)?
Response 1. Body composition measurements were performed using the Body Composition Monitor in accordance with standardized procedures and manufacturer recommendations. All measurements were conducted by trained personnel, with patients in the supine position after a short resting period and bladder voiding, and prior to the main meal of the day. The BCM provides validated estimates of adipose tissue mass, lean tissue mass, and hydration status based on multifrequency bioimpedance spectroscopy, ensuring good reproducibility. Following a thorough re-evaluation of the study design and objectives, we decided not to refer to the Subjective Global Assessment (SGA) in the revised manuscript. This decision was made to maintain methodological coherence and to focus exclusively on objective, repeatable bioimpedance-based measurements.
Comments 2 . Is there a plan for longer term follow-up to monitor nutritional status post-hospitalization and evaluate the persistence of these effects?
Response 2. A structured long-term follow-up was not part of the present study design, which focused on the acute hospitalization period. However, we agree that evaluating post-discharge nutritional trajectories would offer important insights, particularly regarding the persistence or reversibility of observed changes. This has been acknowledged as a limitation and identified as a priority for future research.
Reviewer 3 Report
Comments and Suggestions for Authors
I consider this article, "Beyond Biochemical Markers: Identifying Predictors of Malnutrition in COVID-19," to be relevant, and I congratulate the authors on the topic, which is always important and timeless, as well as the curious and interesting results regarding body composition, such as the extensive loss of adipose tissue, especially in the absence of lean tissue recovery, which likely reflects metabolic exhaustion rather than effective adaptation. Thus, the dynamics of fat mass can provide a sensitive indicator of nutritional decline in patients with COVID-19;
I therefore make some comments that I believe can contribute to the improvement of this article: I agree with the study's limitations pointed out,
I consider Table 1 a bit confusing because the data is not uniformly presented;
Introduction
Paragraph 55………. They only do nutritional screening, they identify nutritional risk
Paragraph 63………. I agree because edema and other aspects can influence the results
Data collection
Paragraph 100………. Why weight loss > 3%?
Paragraph 125………. Perhaps the loss of lean tissue was not measured because the hospitalization time was short, 10 days;
Discussion
Paragraph 181………. Probably the tools, because this study did not use tools to identify nutritional risk;
Paragraph 224………. In your study, was the phase angle not evaluated? But bioimpedance spectroscopy allows this determination, it would have been interesting;
Paragraph 227………. Remember that low phosphorus is related to refeeding syndrome in malnourished patients, which is fundamental to its monitoring.
Author Response
Comments 1. I consider Table 1 a bit confusing because the data is not uniformly presented;
Response 1. We have revised the table to improve clarity and consistency. While the statistical descriptors (mean ± SD or median [IQR]) still vary depending on data distribution, we have clarified this approach in an expanded footnote and ensured that each variable is presented using the most appropriate measure of central tendency. Additionally, we introduced clearer subgrouping (biochemical vs. body composition parameters), standardized unit placement, and adjusted the layout to enhance readability.
Comments 2. Introduction
Paragraph 55………. They only do nutritional screening, they identify nutritional risk
Paragraph 63………. I agree because edema and other aspects can influence the results
Data collection
Paragraph 100………. Why weight loss > 3%?
Response 2. We fully acknowledge that this threshold is not part of any formally validated diagnostic framework. Given that the median length of stay in our cohort was 10 days, we opted for a pragmatic, exploratory threshold to detect early and potentially meaningful nutritional decline. This choice enabled us to identify patients with relevant adipose tissue mass mass loss and supported the development of a predictive model with strong discrimination (AUC 0.85). Nonetheless, we fully recognize the non-standard and arbitrary nature of this threshold and now clarify this point explicitly in the “Limitations” section (lines 300-307). We agree that caution is needed when comparing our findings with studies that use validated diagnostic frameworks.
Comments 3. Paragraph 125………. Perhaps the loss of lean tissue was not measured because the hospitalization time was short, 10 days;
Response 3. We fully agree that the lack of significant lean tissue loss observed in our cohort may be partly related to the relatively short median hospital stay (10 days). This brief period may not have allowed for measurable depletion of lean mass, particularly in patients who were not critically ill and remained at least partially mobile. It is also possible that our findings reflect an early phase of adipose tissue mobilization, which may be more prominent during short-term metabolic stress typically seen in acute inflammatory conditions. Although the literature more frequently describes rapid lean tissue loss in critically ill or immobilized patients, especially in ICU settings (e.g., Puthucheary et al., JAMA 2013), comparative data from non-ICU populations remain limited. We have incorporated this interpretation into the revised Discussion section, as it may help generate hypotheses for future research in broader clinical settings (lines 258-267).
Comments 4. Paragraph 181………. Probably the tools, because this study did not use tools to identify nutritional risk;
Response 4. It is correct that we did not use standardized tools such as NRS-2002 or SGA, which may limit comparability with other studies. Instead, we focused on objective, longitudinal changes in body composition and laboratory markers to capture early nutritional deterioration. While this approach offers certain advantages, we acknowledge that the absence of validated screening tools is a limitation and have clarified this in the revised manuscript (lines 301-302).
Comments 5. Paragraph 224………. In your study, was the phase angle not evaluated? But bioimpedance spectroscopy allows this determination, it would have been interesting;
Response 5. We agree that phase angle (PA) is a valuable parameter reflecting nutritional and cellular status, and its inclusion could have provided additional insight. However, due to technical limitations of the device used in this study (Body Composition Monitor, BCM, Fresenius Medical Care), access to raw impedance data (resistance and reactance) was not available, and therefore PA could not be calculated. In light of this limitation, we decided to remove the discussion section referring to phase angle from the revised manuscript to ensure consistency with the data obtained.
Comments 6. Paragraph 227………. Remember that low phosphorus is related to refeeding syndrome in malnourished patients, which is fundamental to its monitoring.
Response 6. We agree that refeeding syndrome may contribute to phosphorus shifts, particularly in malnourished patients during early nutritional recovery. While we did not formally assess this condition, we have now acknowledged its possible role in the revised “Discussion” to better reflect the complexity of phosphorus dynamics in this context (lines 280-290).
Round 2
Reviewer 1 Report
Comments and Suggestions for Authors
The authors improved the manuscript even if some methodological issues (e.g. the lack of a power analysis) still affect the reliability and interpretability of its findings.